# Chronic cholecystitis: Diagnostic and therapeutic insights from formerly bile-farmed Asiatic black bears (*Ursus thibetanus*)

**Szilvia K. Kalogeropoulu**[1,2]*, **Emily J. Lloyd**[2], **Hanna Rauch**[1,3], **Irene Redtenbacher**[4], **Michael Häfner**[5], **Iwan A. Burgener**[1], **Johanna Painer-Gigler**[3]

**1** Department for Companion Animals and Horses, Division of Small Animal Internal Medicine, University of Veterinary Medicine, Vienna, Austria, **2** Bear Sanctuary Ninh Binh, FOUR PAWS Viet, Ninh Binh, Vietnam, **3** Department of Interdisciplinary Life Sciences, Research Institute of Wildlife Ecology, University of Veterinary Medicine, Vienna, Austria, **4** Wild Animal Department, FOUR PAWS International, Vienna, Austria, **5** Vienna, Austria

* kalogsylvia@gmail.com

**Data Availability Statement:** All relevant data are within the paper and its Supporting information files.

## Abstract

Across Southeast Asia and China, more than 17000 Asian bears are kept under suboptimal conditions and farmed for their bile to meet the consumer demand for traditional medicine products. Years of unsterile and repetitive bile extraction contribute to the development of chronic sterile or bacterial cholecystitis, a pathology commonly diagnosed in formerly bile-farmed bears. In both human and veterinary medicine, the diagnostic value of the macroscopic bile examination for assessing gallbladder disease is unclear. The objective of this study is to identify the role of gallbladder bile color, viscosity, and turbidity, while comparing them with established markers of cholecystitis. Moreover, it aims to define the optimal duration of oral antibiotic treatment for chronic bacterial cholecystitis in bears associated with bile farming. Thirty-nine adult, formerly bile-farmed Asiatic black bears (*Ursus thibetanus*) were examined under anesthesia and underwent percutaneous ultrasound guided cholecystocentesis. A total of 59 bile samples were collected with 20 animals sampled twice to evaluate the therapeutic success. All bile aspirates were assessed macroscopically and microscopically followed by submission for bacterial culture and antimicrobial sensitivity. In the majority of bears, samples with cytological evidence of bactibilia lacked inflammatory cells and did not always correlate with positive bacterial cultures. The most common bacterial isolates were *Enterococcus spp*, *Streptococcus spp* and *Escherichia coli*. Based on our findings, the optimal duration of antibiotic treatment for chronic bacterial cholecystitis is 30 days. Moreover, unlike Gamma-glutamyl Transferase (GGT) and gallbladder wall thickness, the organoleptic properties of bile were found to be reliable markers of chronic gallbladder inflammation with color and turbidity indicating cholestasis. The current study highlights the importance of cholecystocentesis for the management of gallbladder disease and provides initial results on the possible diagnostic value of macroscopic bile examination.

**Funding:** The author(s) received no specific funding for this work.

**Competing interests:** The authors have declared that no competing interests exist.

## Introduction

Bear bile entered the Chinese pharmacopoeia 3000 years ago [1] to treat various ailments [2]. Bile was obtained by killing wild bears and harvesting their gallbladder. In the late 1970s bile extraction techniques from live bears and bear farming practices were developed in Asia [3] to prevent the loss of wild bears and provide an easy and steady supply of bear bile [4]. However, the growth of bear farms contributed to the increase in numbers of bears illegally sourced from the wild or internationally imported [5]. Currently, more than 17,000 bears are estimated to live on farms, under suboptimal conditions, in Southeast Asia and China [3, 6]. The Asiatic black bear (*Ursus thibetanus*) is the most commonly used species in farming operations [3, 7] and is classified as vulnerable with a decreasing population [3].

Bile extraction may be performed daily, or every few weeks, depending on the method and demand and is carried out over the lifespan of an individual. In Vietnam, bile is extracted through unsterile ultrasound guided or blind percutaneous cholecystocentesis [8]. Chronic cholecystitis is the most identified pathology in formerly bile-farmed bears [9]. Gallbladder inflammation with or without bacterial infection is caused due to the unsanitary and traumatic bile extraction method (S1 Fig). Specifically, after bile extraction, the irritated gallbladder wall in response to prostaglandin E2 produces and secretes mucin from its epithelium [10]. Additionally, percutaneously introduced bacteria that can survive the antimicrobial environment of bile [11] proliferate causing infection. The bacterial species identified to exhibit bile tolerance are the Gram-negative bacteria *Salmonella sp*, *Escherichia coli*, *Vibrio cholerae*, *Campylobacter jejuni* and the Gram-positive bacteria *Enterococcus faecalis*, *Listeria monocytogenes* and *Lactobacillus amylovorus* [11]. In general, Gram-positive bacteria seem to be more sensitive to the deleterious effects of bile (bacterial membrane damage, disturbance of bacterial cell macromolecular stability, oxidative and low pH stress) than Gram-negative [11]. Moreover, the increased mucin production promotes the decrease of gallbladder motility and subsequent bile stasis [12]. Prolonged bile stasis will contribute to gallbladder wall ischemia and may allow bacterial colonization (ascending from the gut) and enhancement of the inflammatory response [10, 13]. The optimal duration of antibiotic treatment for chronic cholecystitis in formerly bile-farmed bears is unknown. In some individuals bactibilia resolved after 14 days of antibiotic treatment, while it persisted in most cases. In small animal medicine a minimum of four weeks of antibiotic treatment is recommended for bacterial cholecystitis [14] due to the fact that orally administered antibiotics produce irregular levels in bile resulting in delayed antibacterial activity [15].

The chronicity of the gallbladder inflammation predisposes to cholestasis [16]. The formed mucin strands, cholesterol liquid and monohydrate crystals, bilirubin and calcium salt precipitates will increase the macroscopic turbidity of gallbladder bile and will form sludge or gallstones in moderate to severe cases [17]. We hypothesize that cholestatic gallbladder bile will have a darker color and that dark nuances will positively correlate with macroscopic turbidity.

The objective of this study is to identify the role of the organoleptic properties of bile (color, viscosity and turbidity) while investigating the relationship between these macroscopic characteristics, gallbladder inflammation, and infectious diagnostic markers. Furthermore, we intended to establish the optimal duration of orally administered antibiotic treatment for chronic bacterial cholecystitis in formerly bile-farmed Asiatic black bears.

## Materials and methods

### Ethics statement

All treatments and sampling were part of essential medical interventions. The collection of data during medical procedures was approved (written consent) by the institutional ethics and

animal welfare committee of FOUR PAWS Viet in accordance with the guidelines for good scientific practice from the University of Veterinary Medicine, Vienna and FOUR PAWS International.

## Animals

All animals included in this study were rescued from private bile farms across Vietnam. The majority of the bears were used for bile extraction purposes for a minimum of twelve years, with the exception of four confiscated individuals that were farmed for two to three years. The bears were transferred and housed permanently at BEAR SANCTUARY Ninh Binh (an animal welfare project by FOUR PAWS) in Northern Vietnam.

Thirty-nine (21 female and 18 male), adult, formerly bile-farmed Asiatic black bears (*Ursus thibetanus*) were examined under anesthesia. A total of 59 bile samples were collected with 20 animals sampled twice to evaluate the therapeutic success.

## General health exam

The general health exam included a physical examination, one ventrodorsal and two laterolateral thoracic radiographs, abdominal ultrasound, echocardiography and indirect ophthalmoscopy. Samples collected were blood (20–40 minutes post darting) for complete blood count (mindray BC-2800 Vet Auto Hematology Analyzer, China) and biochemistry (IDEXX VetTest 8008 Chemistry Analyzer, Germany; dry-slide technology), urine for refractometry and culture if indicated, and bile (40–60 minutes post darting) for macroscopic and microscopic examination, bacterial culture and antimicrobial sensitivity. Gamma-glutamyl Transferase (GGT) and all other biochemical parameters were detected and quantified by colorimetric analysis.

The bears were anesthetized with 2.2mg/kg tiletamine-zolazepam (Zoletil 50®, Virbac, Carros, France) + 0.035mg/kg medetomidine (medetomidine 20mg/ml, magistral formulation, Vienna, Austria) + 0.05mg/kg butorphanol (Alvegesic® vet. 10mg/ml, Vienna, Austria), administered intramuscularly via remote projectile. Anesthesia maintenance after 70 minutes from darting was achieved either by intravenous or intramuscular ketamine (1mg/kg ketamine Ket-A-100®, Agrovet Market, Peru) and medetomidine (0.0175mg/kg medetomidine Narcostart® 1mg/ml, Richter pharma, Wels, Austria)(Group I = 22 individuals) every 35 and 80 minutes respectively or by a continuous rate infusion (CRI) with 18mcg/kg/minute of propofol (Fresofol® 1% MCT/LCT, Fresenius-Kabi, Australia) (Group II = 17 individuals) increased to 36mcg/kg/minute at 150 minutes post darting.

Bears with positive bile cultures were treated in most cases according to antibiogram for 14 or 30 days. In case of bacterial resistance to the available agents of the antibiogram, antibiotics were chosen following the recommendations from the human and veterinary medicine literature for the treatment of cholecystitis (bacteria specific) [10, 14, 18–20]. Similarly, cholecystitis treatment duration was decided based on the existing treatment guidelines in other species and the preliminary follow-up gallbladder bile culture results.

All bears received 60 days of oral choleretic treatment consisting of 5mg/kg ursodeoxycholic acid (UDCA) BID, 15mg/kg silymarin SID, 1000mg of artichoke leaf extract SID and 1000mg of *Curcuma comosa* SID and were re-examined 60 days later. For animals with positive bile cultures, bile was recollected and submitted for bacterial culture and sensitivity to assess treatment efficacy.

## Hepatobiliary ultrasonography

Abdominal ultrasound (MyLab™ One Vet- Esaote; SC3421 VET convex probe; 1–8 MHz, The Netherlands) was performed in all 39 bears. Four gallbladder wall measurements were

obtained from the anterior, posterior, left lateral and right lateral wall and a mean value was calculated representing the gallbladder wall thickness. Measurements greater than 1mm were considered as increased wall thickness (based on comparison with five Asiatic black bears free of cholecystitis). Moreover, the gallbladder was assessed for its wall echogenicity, presence of wall edema and contents (i.e., sediment, sludge and gallstones). Additionally, when identified, the common bile duct was measured, and the liver was evaluated for its echogenicity and general morphology.

## Percutaneous transhepatic ultrasound guided cholecystocentesis

Preanesthetic fasting of 12 to 16 hours was implemented. At the beginning of anesthesia, the bears were injected with 0.2mg/kg meloxicam (Metacam® 20mg/ml, Boehringer Ingelheim) SC and continued with per os 5mg/kg firocoxib (Previcox® 227mg tab, Merial) SID for 5 days post cholecystocentesis to suppress local inflammation. Under general anesthesia, all bears were placed in dorsal recumbency and the right upper abdominal quadrant was aseptically prepared for the aspiration. The ultrasound transducer was disinfected after the localization of the gallbladder. A 20G x $2^{3/4}$" sterile needle was attached to an IV extension line (Heidelberger, B| BRAUN, Germany) integrated with a 3-way adapter (Discofix®,B|BRAUN, Germany) and a 10ml syringe. The needle was inserted into the gallbladder under ultrasound guidance through a right-sided transhepatic approach including at least 2cm of liver tissue between the abdominal wall and the anterior gallbladder wall [21–24] to prevent leakage into the abdomen. The volume of bile aspirated ranged between 6 to 10ml.

## Bile examination

**Macroscopic examination.** All 59 bile samples were assessed for their color, viscosity, and turbidity. The macroscopic assessment was performed by one examiner (SK Kalogeropoulu). Specifically, color ranging from light amber to amber was characterized as "amber", "brown" from dark amber to brown and "black" from dark brown to black (S2 Fig). Viscosity was categorized as "mild", "moderate" and "severe" with "mild" considered as the normal macroscopic viscosity of gallbladder bile (x 2 the relative viscosity of water) [12, 25]. Additionally, turbidity was classified as "non-turbid", "mild" and "moderate".

**Cytology.** Direct smears and cytocentrifuge preparations (1000g for 20minutes; [26]) of the bile were prepared within two hours of sampling and stained using Diff-Quik stain (LT-SYS®, Labor+Technik, Germany).

**Bacterial culture.** The bile specimens were cultured aerobically on trypticase soy supplemented with 5% sheep blood and chocolate agar. The cultures were incubated in 5% $CO_2$ at 36°C and were examined for growth daily for 5 days.

**Antimicrobial susceptibility testing.** Antimicrobial susceptibility testing was carried out by using an automated microbiology system according to the manufacturer instructions. Interpretations of susceptible, intermediate or resistant were made according to the breakpoints assigned by the National Committee for Clinical and Laboratory Standards [27].

## Statistical analysis

Statistical analysis was performed by using the software package R for Mac OS X/Windows (R Foundation for Statistical Computing, Vienna, Austria, 2020). A one-way ANOVA test was carried out between bile organoleptic properties (color, viscosity, and turbidity), plasma GGT, and mean gallbladder wall thickness, followed by a Tukey HSD test for multiple comparison of means. The same statistical tests were used to assess the relationship of GGT, mean gallbladder wall thickness, and positive or negative bile cultures. All models were checked for normality

through visual assessment of model assumptions (model residual QQ plots for homoscedasticity and histograms for normal distribution) and a Shapiro-Wilk normality test. Furthermore, the effect of bactibilia on the organoleptic properties of bile was evaluated with a Pearson's Chi-squared test and the effect of treatment duration on treatment efficacy with a Kruskal-Wallis test with pairwise comparisons, due to normality violation. Additionally, a Pearson's Chi-squared test was used to investigate if there is a statistically significant association between the three gallbladder bile colors and levels of turbidity, followed by Bonferroni post- hoc analysis for groupwise comparisons. All results were plotted with ggplot2 package boxplots.

## Results

### General health exam

All animals were diagnosed with chronic sterile or bacterial cholecystitis and chronic hepatopathy of varying severity. Their mean and standard deviation values of gallbladder wall thickness and GGT were 2.91+/-0.96 mm and 105.89+/-2.91IU/L (108.8–102.98) respectively.

### Bile examination

A total of 59 bile samples obtained from 39 Asiatic black bears were examined for their organoleptic properties. The GGT and gallbladder wall thickness means were compared between the categories of organoleptic properties (color, viscosity, turbidity) and all differences were found to be insignificant (p>0.05) (S3–S5 Figs). Moreover, the categorical bile macroscopic characteristics did not differ between bacteria positive and negative bile samples. Specifically, for the comparison of color, viscosity and turbidity with bactibilia, the $\chi^2$ and p-value were equal to 3.93 (df = 2) and 0.14, 2.17 (df = 2) and 0.33, and 0.78 (df = 2) and 0.68, respectively. The relationship of gallbladder bile color and turbidity was found to be statistically significant between "black" color and "non-turbid" turbidity (negative relationship) with $\chi^2$ and p-value equal to15.95 and 0.003 (df = 4).

### Cytology

All 59 bile aspirates were microscopically examined and had a grey background with amorphous aggregates of purple to black material (Fig 1). Golden brown pigment (Fig 2) was present in 8 samples and was interpreted as bilirubin. Microscopic evidence of bactibilia (Fig 2) was shown in 41 cases. Bacterial populations appeared as cocci (N = 25), mixed cocci with bacilli (N = 11) and as bacilli (N = 5). Cholesterol crystals (Fig 3) and hepatobiliary cells (Fig 4) were found in 5 out of 59 bile aspirates. White blood cells (lymphocytes or degenerated neutrophils; Fig 5) were identified in 17 samples and in all cases were consistent with bactibilia. Red blood cells (Figs 4 and 5) were found in one sample out of 59 which was obtained from an individual with chronic calculous cholecystitis with gallstone size equal to 4.45x2.28cm.

### Bacterial culture

Aerobic culture results are presented in Table 1. The most prevalent bacterial genera identified were *Enterococcus*, *Streptococcus* spp and *Escherichia coli*. Furthermore, the comparison of gallbladder wall thickness and GGT means between bacteria positive and bacteria negative bile cultures obtained from chronic cholecystitis patients was insignificant, with p-values equal to 0.39 and 0.15 respectively (S6 Fig).

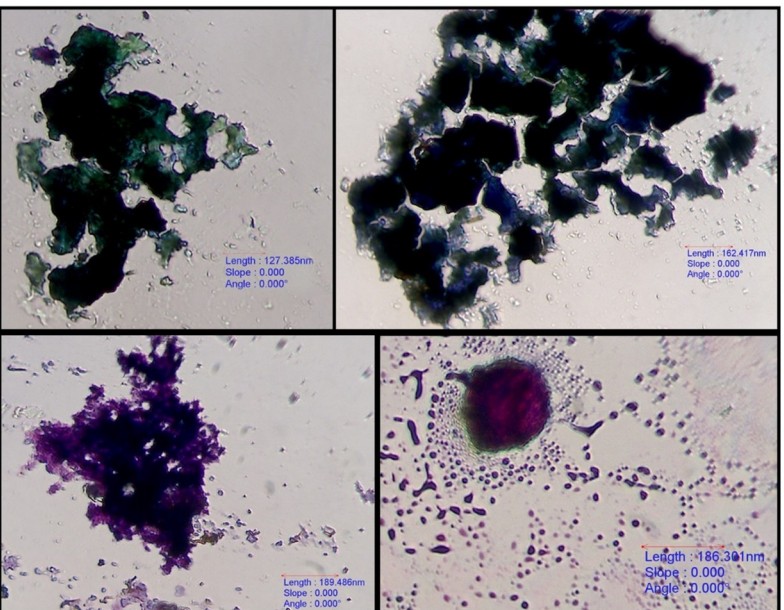

**Fig 1. Purple to black amorphous aggregates.** 40x. Diff-Quik stain, euromex CMEX 5 microscope camera.

## Antimicrobial susceptibility testing

Tables 2 and 3 present the antimicrobial resistance and susceptibility results for both gram-positive and gram-negative microorganisms cultured from the 59 bile samples.

## Antimicrobial treatment duration

The treatment efficacy was evaluated by bacterial culture of bile samples collected following 14 or 30-day antibiotic treatment. Treatment was characterized as effective when culture was negative and ineffective when it was positive. The Kruskal-Wallis test showed a statistically significant difference in treatment efficacy between the different treatment durations of 14 and 30 days, $\chi^2(2) = 10.171$ (df = 2), p = 0.006, with mean rank treatment efficacy score of 2.28 for non-effective and 0.75 for effective treatment (Fig 6).

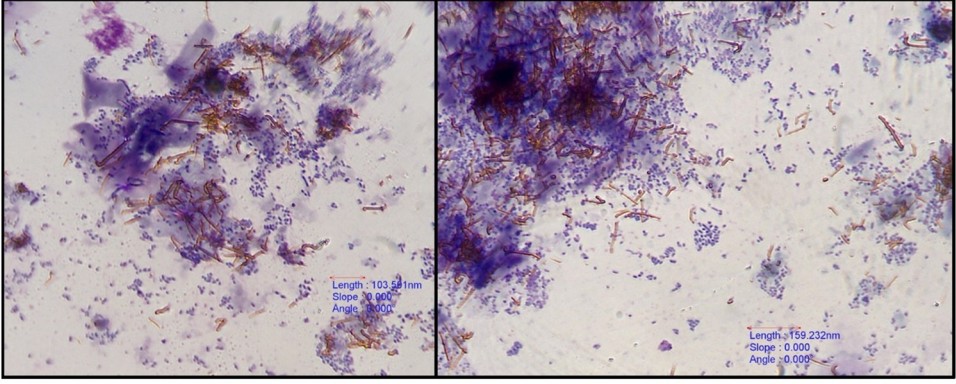

**Fig 2. Bacteria mixed with golden-brown bile pigment.** 40x. Diff-Quik stain, euromex CMEX 5 microscope camera.

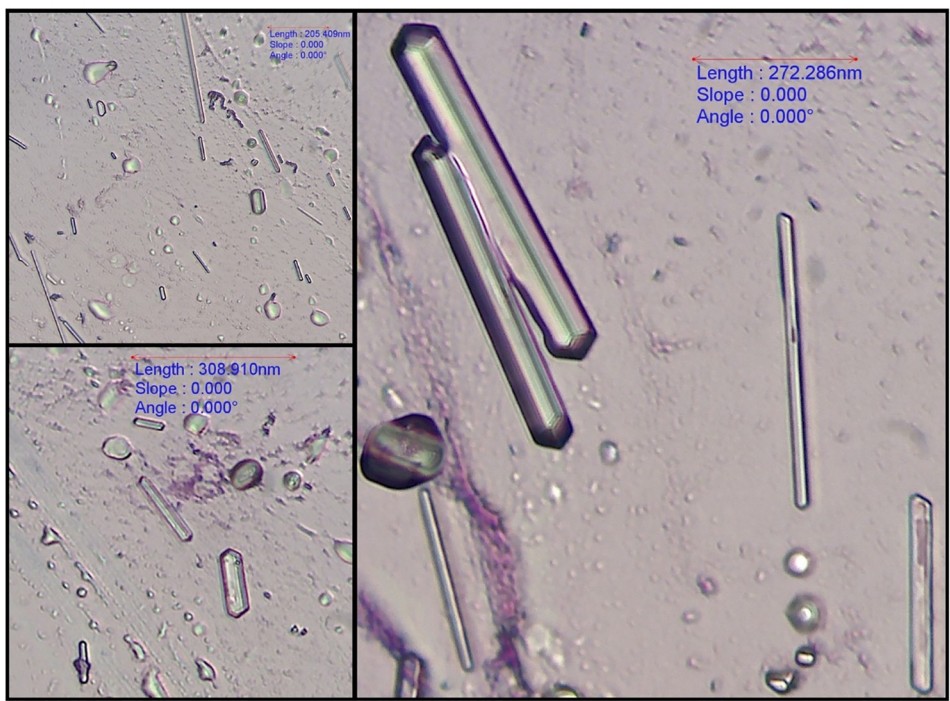

**Fig 3. Cholesterol crystals.** 40x. Diff-Quik stain, euromex CMEX 5 microscope camera.

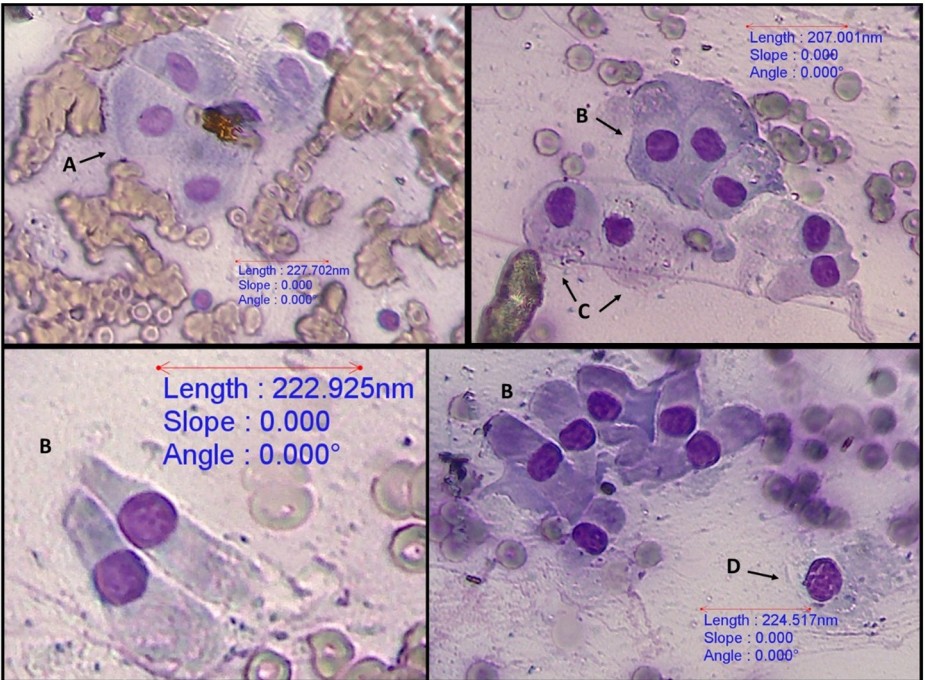

**Fig 4. Red blood cells and hepatobiliary cells.** 40x, (A) hepatocytes, (B) biliary columnar epithelial cells, (C) mesothelial cells, and (D) cuboidal epithelial cell. Diff-Quik stain, euromex CMEX 5 microscope camera.

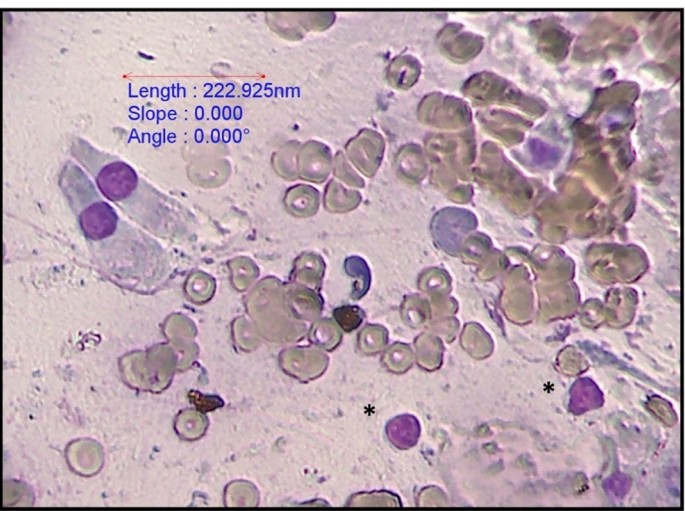

**Fig 5. Biliary columnar cells, red blood cells and small lymphocytes** (*). 40x. Diff-Quik stain, euromex CMEX 5 microscope camera.

Antibiotic treatment duration of 14 and 30 days was compared between Asiatic black bears with positive and negative bile cultures (number of Asiatic black bears in each group is depicted). The comparison (Kruskal-Wallis test with pairwise comparisons) was found to be statistically significant with p-value = 0.006.

## Discussion

Gamma-glutamyl transferase is a liver enzyme that binds to the cell membrane of the hepato-cytes [28] and the epithelial cells of the bile ducts [29]. Its clinical applications are mainly for

**Table 1. Aerobic culture results.**

| | Number (%) of Bacterial Cultures |
|---|---|
| **Bacterial Identity** | *Total* |
| Positive bile cultures | 30/59 (50.8%) |
| *Gram-negative aerobes (18.6%)* | |
| *Escherichia coli* | 6/59 (10.2%) |
| *Pseudomonas aeruginosa* | 3/59 (5%) |
| *Acinetobacter spp* | 1/59 (1.7%) |
| *Klebsiella pneumoniae* | 1/59 (1.7%) |
| *Gram-positive aerobes (32.2%)* | |
| *Enterococcus spp* | 12/59 (20.3%) |
| *E. faecium* | 1/12 |
| *E. faecalis* | 1/12 |
| *E. casseliflavus* | 1/12 |
| *Streptococcus spp* | 6/59 (10.2%) |
| *Staphylococcus spp* | 1/59 (1.7%) |

Identity and prevalence of bacteria isolates by culture of bile from formerly bile-farmed Asiatic black bears (*Ursus thibetanus*) with suspected bacterial chronic cholecystitis.

**Table 2. Antibiogram results for the Gram-negative cultured bacteria.**

| Microorganism | % Resistant / % Susceptible (no. isolates tested) | | | | | | | | | | | | | |
|---|---|---|---|---|---|---|---|---|---|---|---|---|---|---|
| | A | A/C | D | E | IM | ME | CFT | CFZ | AM | NOR | CIP | ENR | COL | NIT |
| *Escherichia coli* | 100%R (3) | 20%R 40%S 40% I (5) | 100%S (2) | 100%S (4) | 100%S (3) | 100%S (3) | 100%S (1) | 66.6%R 33.3%S (3) | 66.6%S 33.3%I (3) | 75%S 25%I (4) | 80%S 20%I (5) | NT | NT | 100%S (3) |
| *Pseudomonas aeruginosa* | 100%R (1) | 100%R (1) | 100%S (2) | NT | 50%R 50%S (2) | 100%S (2) | 100%R (1) | 100%S (2) | 100%S (1) | 100%S (2) | 100%S (3) | NT | 100%S (3) | NT |
| *Acinetobacter spp* | 100%S (1) | 100%S (1) | NT | NT | NT | NT | 100%I (1) | NT | NT | 100%S (1) | | 100%S (1) | 100%S (1) | NT |
| *Klebsiella pneumoniae* | 100%R (1) | 100%I (1) | NT | 100%S (1) | 100%S (1) | 100%S (1) | 100%R (1) | NT | 100%S (1) | 100%S (1) | 100%I (1) | NT | NT | 100%I (1) |

A, ampicillin; A/C, amoxicillin/clavulanate; D, doripenem; E, ertapenem; IM, imipenem; ME, meropenem; CFT, ceftiofur; CFZ, ceftazidime; AM, amikacin; NOR, norfloxacin; CIP, ciprofloxacin; ENR, enrofloxacin; COL, colistin; NIT, nitrofurantoin; NT, not tested; R, resistant; S, sensitive; I, intermediate.

hepatobiliary disorders [30, 31]. GGT levels increase due to impaired bile flow, biliary inflammation [32] or necrosis and cholestasis in some species [33]. The cholestatic sensitivity of serum GGT in dogs is more than 90% [33]. However, the half-life of GGT is approximately 3 days in dogs [31], while serum GGT remained almost stable or decreased after 10 days in experimental bile duct ligation in the same species (cholestasis) [30]. In the current study, the majority of animals had GGT values within reference range with the ISIS physiological intervals for captive Asiatic black bears (17–454 IU/L) [34].

The ultrasonographic imaging finding of thickening of the gallbladder wall is considered characteristic of cholecystitis in small animal and human medicine [32, 35, 36]. Unlike gallbladder wall changes described in acute cholecystitis of humans [36] and small animals [37–39], in formerly bile-farmed bears the gallbladder wall is non-uniformly and mildly to moderately thickened due to the chronicity of the disease and the reoccurring focal wall inflammation related to bile extraction. Gallbladder bile viscosity is a marker of cholecystitis since it increases in pathological cases and positively correlates with the gallbladder bile mucin concentration [12]. Macroscopic gallbladder bile turbidity indicates cholestasis [17] and the statistically significant result of the inverse relationship between presence of turbidity and black color therefore suggests that black gallbladder bile color is an indicator of cholestasis. The differences of GGT and gallbladder wall thickness between the organoleptic property levels or categories were insignificant most likely due the chronicity of the hepatobiliary inflammation.

**Table 3. Antibiogram results for the Gram-positive cultured bacteria.**

| Microorganism | % Resistant / % Susceptible (no. isolates tested) | | | | | | | | | | | |
|---|---|---|---|---|---|---|---|---|---|---|---|---|
| | A | A/C | IM | CFT | CHL | VAN | RIM | TET | NOR | CIP | ENR | NIT |
| *Enterococcus spp* | 82%R | 87.5%R | 88.9%R | NT | 10%R | 16.7%R | 10%R | 63.6%R | 100%R | 81.8%R | NT | 25%R |
| | 18%S | 12.5%S | 11.1%S | | 50%S | | 50%S | 18.2%S | | 18.2%S | | 50%S |
| | | | | | 40%I | 83.3%S | 40%I | 18.2%I | | | | 25%I |
| | (11) | (8) | (9) | | (10) | (12) | (10) | (11) | (10) | (11) | | (12) |
| *Streptococcus spp* | 83.4%R | 100%R | NT | 100%R | 100%R | 100%S | NT | 100%R | 100%R | NT | 100%R | NT |
| | 16.6%S (6) | (5) | | (5) | (1) | (1) | | (1) | (4) | | (4) | |
| *Staphylococcus spp* | NT | 100%R (1) | NT | 100%R (1) | 100%R (1) | NT | NT | 100%R (1) | 100%R (1) | NT | 100%R (1) | NT |

A, ampicillin; A/C, amoxicillin/clavulanate; IM, imipenem; CFT, ceftiofur; CHL, chloramphenicol; VAN, vancomycin; RIM, rifampicin; TET, tetracycline; NOR, norfloxacin; CIP, ciprofloxacin; ENR, enrofloxacin; NIT, nitrofurantoin; NT, not tested; R, resistant; S, sensitive; I, intermediate.

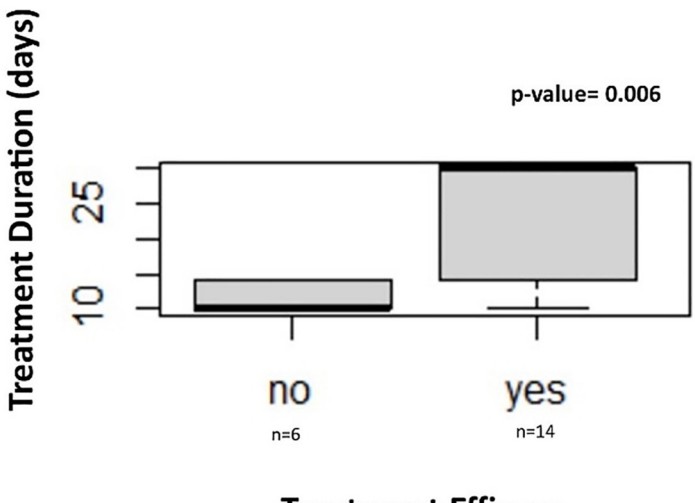

**Fig 6. Antibiotic treatment duration and efficacy for chronic bacterial cholecystitis.**

The majority of the bears with bactibilia lacked cytological evidence of inflammation which may be associated with transient bacterial colonization [40] due to prolonged gallbladder bile stasis. Only one sample collected from a bear with severe cholelithiasis and decreased gallbladder emptying contained red blood cells. It is assumed that the presence of red blood cells is associated with the traumatic effect of the gallstone within the gallbladder. Even though 41 bile aspirates showed cytological evidence of bactibilia, only 30 had a positive bacterial culture. It is possible that the eleven remaining cultures were negative because they were cultured only aerobically (anaerobic cultures were unavailable in local laboratories), or due to the bacteriostatic effect of bile or the difficulty to culture certain bacterial species. From the 41 samples, 11 presented mixed bacterial populations with only one bacterial culture positive for two microorganisms, while the rest of the samples were positive for a single organism. It is presumed that bacterial culture of a single organism is associated with in vitro overgrowth of one bacterial species [40].

The Gram-positive and Gram-negative bacteria isolated from the 59 bile aspirates were consistent with findings reported in humans [19, 20], small animals [39, 40, 41–44], single cases of a kinkajou (*Potos flavus*) [45], and a domestic ferret (*Mustela putorius*) (non-bile farmed) [46], despite the additional infection pathway related to bile extraction. Additionally, in formerly bile-farmed bears rescued from Chinese bile farms the bacteria isolated were *E. coli*, *Enterococcus spp* and *Pseudomonas spp* [9]. In China, bile is extracted continuously by surgically created transabdominal gallbladder fistulas or indwelling catheters [47]. It appears that, independently of the pathogenesis mechanism of cholecystitis and bactibilia, only certain bacterial Genera can survive the bacteriostatic effect of bile. Furthermore, in the current study the bacteria predominantly isolated were Gram-positive aerobes. This finding is not consistent with Begley et al., who suggested that Gram-positive bacteria are more sensitive to the deleterious effects of bile than Gram- negative. The antimicrobial effect of bile is primarily achieved through the bile acids, which alter the membrane integrity and permeability of bacterial cells by interacting with their membrane lipids and causing membrane damage [11]. We hypothesize that chronic gallbladder hypomotility contributes to reduced gallbladder bile acid pool [48, 49] resulting in decreased antimicrobial efficacy of bile and allowing bacteria, that would

normally not survive, to proliferate in the gallbladder. The insignificant effect of positive bacterial cultures on GGT and gallbladder wall thickness is most likely also related to the chronic inflammation.

In our study, the majority of the most commonly isolated Gram-positive microorganisms were resistant to beta-lactams (>82%), with the exception of two samples. Specifically, *Enterococcus faecalis* isolated from one of the samples, was susceptible to ampicillin and to amoxicillin-clavulanate while, *Streptococcus spp* from the other sample was sensitive to ampicillin. Moreover, *Enterococcus spp* and *Streptococcus spp* were also resistant to fluoroquinolones (>81%) and tetracycline (63.6% and 100%, respectively). Vietnam is considered a potential hot-spot for the emergence of antimicrobial drug resistance (AMR) [50]. In human health, the predisposing factors for AMR are inadequate laboratory instrumentation and insufficiently trained staff for bacterial culture and antibiogram in many hospitals across the country. Moreover, market liberation and privatization of health care have resulted in large portions of the population being unable to afford healthcare services, as well as 'over the counter' antibiotic availability and financial benefits for the antibiotic prescriber since the majority of the antibiotics are made locally [50–52]. Consequently, self-medicating with antibiotics often in low dosages and prescription of inappropriate antimicrobial agents take place. In animal production, antibiotics are used by farmers excessively without veterinary supervision either "prophylactically" due to the low farm hygiene standards or as growth promoters to increase productivity [53]. Specifically in pig and poultry farms aminoglycosides, beta-lactams, phenicols, tetracyclines, fluoroquinolones, sulphonamides, ionophores and colistin are often utilized [54–56]. In bear bile farming antibiotics are also used "prophylactically" without veterinary supervision and in incorrect dosages and duration to "prevent" further infectious development after unsterile bile extraction. During rescue missions, empty vials of beta-lactams such as penicillin and ampicillin are often found next to the bear cages. Furthermore, the diet of bile-farmed bears consists partly or entirely of pig feeds with medicated feeds being frequently used. AMR in animals is inherently linked to AMR in humans [51]. Our results suggest that the AMR identified in formerly bile-farmed bears is consistent with the general AMR status in the country.

Bear bile farming remains a common practice in Asia, compromising the health and welfare of at least 17000 Asian bears and is one of the main reasons for Asiatic black bear population decline [3]. Morbidities associated with bear bile farming are understudied, making scientific research on this topic essential. The current study provides novel and comprehensive information for the diagnosis and treatment of hepatobiliary disease in formerly bile-farmed bears, facilitating the health management of rescued individuals in Southeast Asia and China.

To conclude, the optimal duration of antibiotic treatment for chronic bacterial cholecystitis in formerly bile-farmed Asiatic black bears based on our findings and considering the AMR status of Vietnam is 30 days, as also recommended in small animal medicine. Furthermore, despite the fact that gallbladder bile color, turbidity and viscosity are inflammatory markers, with color and turbidity indicating cholestasis, they were not associated with GGT and gallbladder wall thickness due to the chronicity of cholecystitis in these animals. Thereby organoleptic properties of bile could be considered a sensitive diagnostic tool of cholecystitis independently of disease chronicity and may be applicable in a wide range of mammalian species.

## Supporting information

**S1 File. Minimal dataset included in this study.**
(CSV)

**S1 Fig. Pathophysiology of chronic cholecystitis in formerly bile-farmed Asiatic black bears.**
(TIF)

**S2 Fig. Bile color nuances.** The color samples were obtained from photographs of the collected bile samples.
(TIF)

**S3 Fig. Boxplot: Gallbladder wall thickness, GGT and bile color.** Gallbladder wall thickness and GGT means were compared between different bile color categories of Asiatic black bears (number of Asiatic black bears in each group is depicted) with chronic cholecystitis. The comparisons (one-way ANOVA) were found to be statistically insignificant with p-value> 0.05.
(TIF)

**S4 Fig. Boxplot: Gallbladder wall thickness, GGT and bile viscosity.** Gallbladder wall thickness and GGT means were compared between different bile viscosity levels of Asiatic black bears (number of Asiatic black bears in each group is depicted) with chronic cholecystitis. The comparisons (one-way ANOVA) were found to be statistically insignificant with p-value> 0.05.
(TIF)

**S5 Fig. Boxplot: Gallbladder wall thickness, GGT means and bile turbidity.** Gallbladder wall thickness and GGT means were compared between different bile turbidity levels of Asiatic black bears (number of Asiatic black bears in each group is depicted) with chronic cholecystitis. The comparisons (one-way ANOVA) were found to be statistically insignificant with p-value> 0.05.
(TIF)

**S6 Fig. Boxplot: Gallbladder wall thickness, GGT and bactibilia.** Gallbladder wall thickness and GGT means were compared between positive and negative bile cultures of Asiatic black bears (number of Asiatic black bears in each group is depicted) with chronic cholecystitis. The comparisons (one-way ANOVA) were found to be statistically insignificant with p-value> 0.05.
(TIF)

## Acknowledgments

The authors thank IDEXX Austria for donating the VetTest 8008 Chemistry Analyzer for this study and Dr Bonnie L Raphael for revising the text.

## Author Contributions

**Conceptualization:** Szilvia K. Kalogeropoulu.

**Data curation:** Szilvia K. Kalogeropoulu.

**Formal analysis:** Szilvia K. Kalogeropoulu, Hanna Rauch.

**Project administration:** Emily J. Lloyd, Irene Redtenbacher.

**Supervision:** Iwan A. Burgener, Johanna Painer-Gigler.

**Writing – original draft:** Szilvia K. Kalogeropoulu.

**Writing – review & editing:** Michael Häfner, Iwan A. Burgener, Johanna Painer-Gigler.

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
