## [Decision Letter · Decision Letter 0]

5 Oct 2021

PONE-D-21-26096Cholecystocentesis: a valuable diagnostic tool for successful treatment of chronic cholecystitis in formerly bile-farmed Asiatic black bears (Ursus thibetanus).PLOS ONE

Dear Dr. Kalogeropoulu,

Thank you for submitting your manuscript to PLOS ONE. After careful consideration, we feel that it has merit but does not fully meet PLOS ONE’s publication criteria as it currently stands. Therefore, we invite you to submit a revised version of the manuscript that addresses the points raised during the review process.

ACADEMIC EDITOR:I have completed my evaluation of your manuscript. The study seems interesting enough for the readership and meets the publication criteria of the Journal. I recommend reconsideration of the manuscript following major revision. When revising your manuscript, please consider all issues mentioned in the reviewer's comments carefully.

I also suggest the authors revise the manuscript and an Abstract in order to highlight the significance and strength of their research. 

I also regret  this long review period. It was attributed to the incapability to aquire the compliance of invited experts to evaluate your potentially interesting manuscript. An evaluation of the submitted manuscripts is principally dependent on contributory experts and is often hampered by a long search for responding experts. To date, only one of the 14 invited experts has agreed to contribute to a review process. If the authors decide to resubmit the manuscript I also suggest that authors provide potential reviewers who could additionally evaluate your study.

We look forward to receiving your revised manuscript.

Kind regards,

Mariya Y Pakharukova, Ph.D., D.Sc.

Academic Editor

PLOS ONE

Reviewers' comments:

Reviewer's Responses to Questions

**Comments to the Author**

1. Is the manuscript technically sound, and do the data support the conclusions?

Reviewer #1: Yes

2. Has the statistical analysis been performed appropriately and rigorously? 

Reviewer #1: Yes

3. Have the authors made all data underlying the findings in their manuscript fully available?

Reviewer #1: Yes

4. Is the manuscript presented in an intelligible fashion and written in standard English?

Reviewer #1: Yes

5. Review Comments to the Author

Reviewer #1: The manuscript is well written and the study was conducted in rigorous manner. But the only scientific conclusion of the study is the revealing of diagnostic tools and establishing of effective duration of antibiotic therapy of cholecystitis in captured black bears. This is the excellent material for any specialised veterinary journal but not for PLOS ONE that publish only the papers of fundamental scientific importance.

The special notes are as follows:

1. The manuscript is overloaded by the illustrative material. There are 13 figures and three tables in main text. Authors should give the majority of the raw information as supplementary, giving in the text only the most essential data. Box and whiskers on the diagrams should be explained in the legends.

2. The Gamma glutamyl Transferase used in the study as one of the indicators of cholecystitis but no description of the assay is given.

Lines 258-260: "Furthermore, the comparison of gallbladder wall thickness and GGT means between bacteria positive and negative bile cultures was insignificant ...". Was the "comparison" really insignificant? May be difference was?

Lines 285-286: "The Kruskall-Wallis test showed a statistically significant difference in treatment efficacy between the different treatment durations". How the authors estimated the treatment efficacy? It must be clearly explained.

6. PLOS authors have the option to publish the peer review history of their article (what does this mean?). If published, this will include your full peer review and any attached files.

Reviewer #1: No

---

## [Author Response · Author response to Decision Letter 0]

10 Nov 2021

Dear Academic Editor, dear Reviewer,

Thank you very much for considering our publication “Cholecystocentesis: a valuable diagnostic tool for successful treatment of chronic cholecystitis in formerly bile-farmed Asiatic black bears (Ursus thibetanus)”to be reviewed for PLOS ONE. We hope that the new version of our manuscript fulfills the publication requirements of your journal. Please find below the reply to each point raised in the comments. Please also find uploaded a marked-up copy of our revised manuscript entitled “Revised Manuscript with Track Changes”, an unmarked version of our revised paper entitled “Manuscript”, the formatted figures and the supporting information files.

Academic Editor’s comments:

1. Revision of the abstract to highlight the significance and strength of the research. � Answer (A): the changes were done accordingly.

2. Revision of the text (discussion) to highlight the significant and strength of the research. � A: the changes were done accordingly.

3. Provision of potential reviewers. � A: we would like to suggest the following researchers as potential reviewers:

a. Jan S Suchodolski

orcid.org/0000-0002-2176-6932

Email: jsuchodolski@cvm.tamu.edu

Texas A&M University College Station

UNITED STATES

Sections: Veterinary science

Keywords: Biology and life sciences, Microbial ecology, Immunology, Inflammation, Immunity, Clinical immunology, Microbiology, Microbial pathogens, Veterinary science, Veterinary diseases, Veterinary medicine, Small animal care, Veterinary diagnostics, Veterinary microbiology, Biochemistry, Enzymes, Medicine and health sciences, Gastroenterology and hepatology, Infectious diseases, Bacterial diseases, Zoonoses, Leishmaniasis, Pathology and laboratory medicine, Anatomical pathology, Histopathology, Endoscopy, Hepatosplenomegaly, Diagnostic medicine, Retrospective studies, Clinical research design

b. Kristin Mühldorfer

orcid.org/0000-0001-8023-4736

Email: muehldorfer@izw-berlin.de

Leibniz Institute for Zoo and Wildlife Research (IZW)

GERMANY

Sections: Veterinary science, Microbiology - Bacteriology, Infectious diseases - Bacterial and fungal diseases

Keywords: Biology and life sciences, Microbial evolution, Bacterial evolution, Microbial genomics, Bacterial genomics, Bacterial genomes, Microbiome, Bacterial genetics, Microbiology, Bacteriology, Bacterial taxonomy, Gram negative bacteria, Gram positive bacteria, Fungal pathogens, Molecular biology, Molecular biology techniques, Mycology, Veterinary parasitology, Veterinary science, Veterinary diseases, Veterinary medicine, Veterinary diagnostics, Veterinary microbiology, Veterinary pathology, Veterinary virology, Zoonoses, Research and analysis methods

c. Peter Starkel

Email: peter.starkel@uclouvain.be

Cliniques Universitaires Saint-Luc

BELGIUM

Sections: Gastroenterology and hepatology

Keywords: Signal transduction, Signaling cascades, AKT signaling cascade, Primary biliary cirrhosis, Crohn's disease, Medicine and health sciences, Gastroenterology and hepatology, Inflammatory bowel disease, Biliary disorders, Primary sclerosing cholangitis, Hepatitis

d. Maite Garcia Fernández-Barrena

orcid.org/0000-0003-0375-6236

Email: magarfer@unav.es

Centre for Applied Medical Research, University of Navarra

SPAIN

Sections: Gastroenterology and hepatology, Cancer - Genetics and screening, Genetics - Gene expression; Epigenetics; Alternative splicing; RNA splicing; Molecular genetics

Keywords: Biology and life sciences, Biochemistry, Medicine and health sciences, Gastroenterology and hepatology, Inflammatory bowel disease, Hepatocellular

4. Please ensure that your manuscript meets PLOS ONE's style requirements, including those for file naming � A: The manuscript including the file names was formatted according to the PLOS ONE style requirements. 

5. Upload minimal dataset underlying the results- as supporting information file. � A: We added the minimal dataset.

6. Include ethics statement in the material and methods: � A: an ethical statement was given in line 95-99.

Reviewer #1 comments:

Thank you very much for taking the time and reviewing our manuscript. Your input is very much appreciated! Please find below our replies to your points:

1. Manuscript overloaded with illustrative material: � A: We deleted 5 figures and moved them to the supportive information section. Please let us know in case the manuscript still appears overloaded and would like us to delete additional figures. Boxes and whiskers were explained in the legends.

2. Clarification of manuscript’s fundamental scientific importance. � A: We consider our findings of fundamental scientific importance, since they provide novel and comprehensive information for the diagnosis and treatment of hepatobiliary disease in formerly bile farmed bears, an understudied topic of great concern for wildlife veterinarians, conservationists and bear husbandry related professionals in South East Asia and China. Bear bile farming is still a common practice in Asia, despite being the main threat for population decline of Asiatic black bears and compromising the welfare of at least 17000 individuals, making this manuscript an essential contribution. Moreover, the findings presented in the current study may be applicable in other mammalian species and are not limited only to Asiatic black bears. 

3. Provision of description of GGT as an assay: � A: description of the assay was given in line 271.

4. Lines 258-260: "Furthermore, the comparison of gallbladder wall thickness and GGT means between bacteria positive and negative bile cultures was insignificant ...". Was the "comparison" really insignificant? May be difference was? � A: The comparison of GGT and gallbladder wall thickness was statistically insignificant between bacteria positive and bacteria negative bile samples collected from individuals with chronic cholecystitis and both p-values were greater than 0.05. The clarification can be found in the lines 235-238 of the revised manuscript. 

5. Clear explanation of treatment efficacy estimation (lines 285-286, non-revised MS version)�A: The clarification of treatment efficacy estimation is given in lines 259-261.

Other corrections

1. The email of the corresponding author was corrected in line 22.

2. The dose of the continuous rate infusion of propofol was corrected in lines 123-125.

3. Escherichia coli was added as one of the most prevalent bacterial genera identified in the bile cultures in line 235.

4. The supportive information captions were added in lines 519-533.

Thank you very much for re-considering our manuscript to be published after major revision.

Kind regards,

Szilvia K. Kalogeropoulu

---

## [Decision Letter · Decision Letter 1]

6 Dec 2021

PONE-D-21-26096R1Cholecystocentesis: a valuable diagnostic tool for successful treatment of chronic cholecystitis in formerly bile-farmed Asiatic black bears (Ursus thibetanus).PLOS ONE

Dear Dr. Kalogeropoulu,

Thank you for submitting your manuscript to PLOS ONE. After careful consideration, we feel that it has merit but does not fully meet PLOS ONE’s publication criteria as it currently stands. Therefore, we invite you to submit a revised version of the manuscript that addresses the points raised during the review process.

ACADEMIC EDITOR:

The Reviewer claimed a number of corrections and improvements. The authors are invited to address these and resubmit a further improved revised manuscript accompanied by your detailed response to all comments point-by-point, including a description of the changes made in the revised manuscript. 

When revising your manuscript, please consider all issues mentioned in the Reviewer' comments carefully: please outline every change made in response to the comments and provide suitable rebuttals for any comments not addressed. 

We look forward to receiving your revised manuscript.

Kind regards,

Mariya Y Pakharukova, Ph.D., D.Sc.

Academic Editor

PLOS ONE

Reviewers' comments:

Reviewer's Responses to Questions

**Comments to the Author**

1. If the authors have adequately addressed your comments raised in a previous round of review and you feel that this manuscript is now acceptable for publication, you may indicate that here to bypass the “Comments to the Author” section, enter your conflict of interest statement in the “Confidential to Editor” section, and submit your "Accept" recommendation.

Reviewer #1: (No Response)

2. Is the manuscript technically sound, and do the data support the conclusions?

Reviewer #1: Yes

3. Has the statistical analysis been performed appropriately and rigorously? 

Reviewer #1: No

4. Have the authors made all data underlying the findings in their manuscript fully available?

Reviewer #1: Yes

5. Is the manuscript presented in an intelligible fashion and written in standard English?

Reviewer #1: Yes

6. Review Comments to the Author

Reviewer #1: Authors considered most issues of first review and significantly improved the manuscript. However some questions and notes still remain.

1. In reviewed version authors shifted accents towards the fundamental value of the study. But the title still underlie the practical veterinary aspect.May be authors should correct the title and make it more "neutral". For ex., just "Chronic cholecystitis occasion in formerly bile-farmed 3 Asiatic black bears (Ursus thibetanus).

2. In respond to my note concerning the description of GGT assay authors write: "description of the assay was given

in line 271". There is no any description neither in old, nor in new version on this page.

3. How two different parameters - gallbladder bile turbidity and color can be statistically "compared"? To my mind one can analyse either correlation between two quantitative parameters, or qualitative effects of independent variable on dependent one. Respective figure (fig.2) does not clarify this question. If the turbidity was classified on three categories "mild", "moderate", "no", what do the intermediate points (0.5, 1.5) on Y-axe means? Additionally this figure is too flattened to be perceived correctly. Three mentions on statistical significance of the comparision (in text, in caption and on the diagram) seem to be redundant. Instead, authors should explain the box and whiskers values.

4. What is the total time animal's spent under anesthesia?

5. "Bears with positive bile cultures were treated based on antibiogram or received other antibiotics

in cases of bacterial resistance for 14 or 30 days". If I understand correctly, the exact course of antibiotic treatment was chosen personally, basing of the individual susceptibility to different antibiotics. How can the durations and effectiveness of the courses can be compared and optimal duration established? For me and other readers who are not experts in veterinary the optimal duration of the course seems to be strongly depended on the infection agent and medicines applied.

6. Fig.1 contains no essential information and may be omitted.

7. PLOS authors have the option to publish the peer review history of their article (what does this mean?). If published, this will include your full peer review and any attached files.

Reviewer #1: **Yes: **Eugene Novikov

---

## [Author Response · Author response to Decision Letter 1]

18 Jan 2022

Rebuttal letter points – PONE-D-21-26096R1

Dear Academic Editor, dear Reviewer, 

Thank you very much for taking the time and reviewing our manuscript after the first major revision. We hope that this edited version following your recommendations and comments will fulfill the requirements for publication in PLOS ONE. Please see below the reply to each point raised in the comments. Please also find attached a marked-up copy of our revised manuscript entitled “Revised Manuscript with Track Changes”, an unmarked version of our revised paper entitled “Manuscript”, the formatted figures and the supporting information files.

Academic editor’s comments:

The comments of the reviewer were considered carefully and effort was made to answer every point precisely and modify the text accordingly. 

Reviewer comments:

Thank you again for meticulously reviewing our manuscript and helping us to present and communicate better our findings. Your comments truly improved the quality of this manuscript!

1. Title correction and making it more neutral Answer (A): the manuscript’s current title is “Cholecystocentesis: a valuable diagnostic tool for successful treatment of chronic cholecystitis in formerly bile-farmed Asiatic black bears (Ursus thibetanus)”. Our suggested new title is: “Chronic cholecystitis: diagnostic and therapeutic insights from formerly bile-farmed Asiatic black bears (Ursus thibetanus)”. Please find it in lines: 1-3.

2. Provision of description of GGT as an assay�A: Please find GGT assay description in lines 116-117. The GGT analysis was done with the Vettest 8008 machine from IDEXX. The machine uses dry-slide technology (inputted in line 114) and GGT was detected and quantified with colorimetric analysis. We are hoping that our new explanation sufficiently answered your question. 

3. How two different parameters - gallbladder bile turbidity and color can be statistically "compared"? To my mind one can analyse either correlation between two quantitative parameters, or qualitative effects of independent variable on dependent one. Respective figure (fig.2) does not clarify this question. If the turbidity was classified on three categories "mild", "moderate", "no", what do the intermediate points (0.5, 1.5) on Y-axe means? Additionally, this figure is too flattened to be perceived correctly. Three mentions on statistical significance of the comparison (in text, in caption and on the diagram) seem to be redundant. Instead, authors should explain the box and whiskers values� A: Gallbladder bile turbidity and color have been treated as categorical variables with three levels each. A chi-square test can be used to analyze two categorical variables and evaluates whether there is a significant association between the categories of the variables. Usually, box plots help to visualize the distribution of quantitative values in a field. It is recognized that in our case that we have two categorical variables the box plot is not an appropriate visualization method. Moreover, in Figure 2, the categories of turbidity “non-turbid”, “mild” and “moderate” were replaced by “0”, “1” and “2” respectively to allow the visualization of the data as a box-plot and the intermittent points 0.5 and 1.5 did not represent any category of bile turbidity. The Pearson’s Chi- squared test was repeated and followed by a Bonferroni post-hoc analysis for groupwise comparison to identify which categorical pair/s of the variables analyzed was/were significantly associated. The significant p-value of the chi-square test is attributed to the negative association between “black” bile color and “non-turbid” turbidity. Regarding result visualization for small data sets and categorical variables a bar plot is recommended. Despite the statistically significant relationship between “black” and “non-turbid”, this finding was poorly visualized by the bar plot. Bar plots are a descriptive tool that show the distribution of the data and since “amber” was the most common finding led to its “overrepresentation” in the graph suggesting that the relationship with “non-turbid” is statistically significant, while the association of “black” and “non-turbid” appears to be insignificant (please see below). Considering the above points and trying to provide clarity for the reader we would like to remove the former box plot (fig 2) and present our findings only via text. Changes in the “materials and methods” section can be found in lines 195-198. Modifications in the results and discussion are made in lines 212-214 and 296-297 respectively. Thank you for your critical thought, we hope that the statistical changes are conform with you. We believe that your comment has much improved our result communication to the reader!

4. What is the total time animal spent under anesthesia? � A: Data were collected during essential medical interventions suited to each bear’s medical needs and were not part of a designed experiment, hence the total duration of anesthesia varied between individuals. In an effort to standardize sampling, blood samples were collected within 20-40 minutes post darting and bile samples between 40-60 minutes post darting. This information was added to the manuscript text in lines 112 and 115 respectively. 

5. "Bears with positive bile cultures were treated based on antibiogram or received other antibiotics in cases of bacterial resistance for 14 or 30 days". If I understand correctly, the exact course of antibiotic treatment was chosen personally, basing of the individual susceptibility to different antibiotics. How can the durations and effectiveness of the courses can be compared and optimal duration established? For me and other readers who are not experts in veterinary the optimal duration of the course seems to be strongly depended on the infection agent and medicines applied. �A: The exact course of antibiotic treatment was not chosen personally, but according to the existing literature for the treatment of bacterial cholecystitis. From both human and veterinary medicine there are antibiotic treatment recommendations (agent, treatment duration) against specific microorganisms causing cholecystitis. In cases where bears were resistant to the antibiotics tested in the antibiogram, an antibiotic (not included in the antibiogram) was chosen based on the existing treatment guidelines from other species. The clarification was made in lines 128-133. Literature examples of treatment recommendations were cited and the reference list numbering was adjusted accordingly. 

6. Omit Figure 1�A: Figure 1 was omitted and added to supporting information. Figure numbering and naming was changed accordingly. Additionally, the legend of Figure 1 as a supporting figure was given in lines 534-535. 

Thank you very much considering our manuscript for publication after a second major revision. 

Kind Regards, 

Szilvia K. Kalogeropoulu

---

## [Decision Letter · Decision Letter 2]

10 Feb 2022

Chronic cholecystitis: diagnostic and therapeutic insights from formerly bile-farmed Asiatic black bears (Ursus thibetanus).

PONE-D-21-26096R2

Dear Dr. Kalogeropoulu,

We’re pleased to inform you that your manuscript has been judged scientifically suitable for publication and will be formally accepted for publication once it meets all outstanding technical requirements.

Kind regards,

Mariya Y Pakharukova, Ph.D., D.Sc.

Academic Editor

PLOS ONE

Additional Editor Comments (optional):

Reviewers' comments:

Reviewer's Responses to Questions

**Comments to the Author**

1. If the authors have adequately addressed your comments raised in a previous round of review and you feel that this manuscript is now acceptable for publication, you may indicate that here to bypass the “Comments to the Author” section, enter your conflict of interest statement in the “Confidential to Editor” section, and submit your "Accept" recommendation.

Reviewer #1: All comments have been addressed

Reviewer #2: All comments have been addressed

2. Is the manuscript technically sound, and do the data support the conclusions?

Reviewer #1: Yes

Reviewer #2: Yes

3. Has the statistical analysis been performed appropriately and rigorously? 

Reviewer #1: Yes

Reviewer #2: N/A

4. Have the authors made all data underlying the findings in their manuscript fully available?

Reviewer #1: Yes

Reviewer #2: Yes

5. Is the manuscript presented in an intelligible fashion and written in standard English?

Reviewer #1: Yes

Reviewer #2: Yes

6. Review Comments to the Author

Reviewer #1: (No Response)

Reviewer #2: The manuscript presents interesting and important research. The authors were aimed to define the optimal treatment for chronic bacterial cholecystitis in Asiatic black bears (Ursus thibetanus). Moreover, their research was also aimed at identification the role of gallbladder bile color, viscosity, and turbidity, while comparing them with established markers of cholecystitis.

After reviewing the previous versions of the text, one may conclude that the authors changed the text and improved the manuscript significantly.

In this version, the authors provide quite sufficient methodological details of their experiments. The details of suggested approach are described well and the approach look appropriate.

7. PLOS authors have the option to publish the peer review history of their article (what does this mean?). If published, this will include your full peer review and any attached files.

Reviewer #1: No

Reviewer #2: No

---

## [Editor Report · Acceptance letter]

18 Feb 2022

PONE-D-21-26096R2 

Chronic cholecystitis: diagnostic and therapeutic insights from formerly bile-farmed Asiatic black bears (*Ursus thibetanus*). 

Dear Dr. Kalogeropoulu:

I'm pleased to inform you that your manuscript has been deemed suitable for publication in PLOS ONE. Congratulations! Your manuscript is now with our production department. 

Kind regards, 

on behalf of

Dr. Mariya Y Pakharukova 

Academic Editor

PLOS ONE